# Prognostication in Advanced Cancer by Combining Actigraphy-Derived Rest-Activity and Sleep Parameters with Routine Clinical Data: An Exploratory Machine Learning Study

**DOI:** 10.3390/cancers15020503

**Published:** 2023-01-13

**Authors:** Shuchita Dhwiren Patel, Andrew Davies, Emma Laing, Huihai Wu, Jeewaka Mendis, Derk-Jan Dijk

**Affiliations:** 1Department of Clinical and Experimental Medicine, Faculty of Health and Medical Sciences, University of Surrey, Guildford GU2 7XP, UK; 2Trinity College Dublin, University College Dublin and Our Lady’s Hospice, DRW RY72 Dublin, Ireland; 3School of Biosciences, Faculty of Health and Medical Sciences, University of Surrey, Guildford GU2 7XP, UK; 4Surrey Clinical Trials Unit, University of Surrey, Guildford GU2 7XP, UK; 5Surrey Sleep Research Centre, Department of Clinical and Experimental Medicine, Faculty of Health and Medical Sciences, University of Surrey, Guildford GU2 7XP, UK; 6Care Research and Technology Centre, UK Dementia Research Institute, Imperial College London and University of Surrey, Guildford GU2 7XP, UK

**Keywords:** biomarkers, circadian, machine learning, palliative care, prognosis, survival

## Abstract

**Simple Summary:**

Survival prediction is an important aspect of oncology and palliative care. Measures of night-time relative to daytime activity, derived from a motion sensor, have shown promise in patients receiving chemotherapy. Measuring rest-activity and sleep may, therefore, result in improved prognostication in advanced cancer patients. Fifty adult outpatients with advanced cancer were recruited, and rest-activity, sleep, and routine clinical variables were collected just over a one week period, and used in machine learning models. Our findings confirmed the importance of some well-established survival predictors and identified new ones. We found that sleep-wake parameters may be useful in prognostication in advanced cancer patients when combined with routinely collected data.

**Abstract:**

Survival prediction is integral to oncology and palliative care, yet robust prognostic models remain elusive. We assessed the feasibility of combining actigraphy, sleep diary data, and routine clinical parameters to prognosticate. Fifty adult outpatients with advanced cancer and estimated prognosis of <1 year were recruited. Patients were required to wear an Actiwatch^®^ (wrist actigraph) for 8 days, and complete a sleep diary. Univariate and regularised multivariate regression methods were used to identify predictors from 66 variables and construct predictive models of survival. A total of 49 patients completed the study, and 34 patients died within 1 year. Forty-two patients had disrupted rest-activity rhythms (dichotomy index (I < O ≤ 97.5%) but I < O did not have prognostic value in univariate analyses. The Lasso regularised derived algorithm was optimal and able to differentiate participants with shorter/longer survival (log rank *p* < 0.0001). Predictors associated with increased survival time were: time of awakening sleep efficiency, subjective sleep quality, clinician’s estimate of survival and global health status score, and haemoglobin. A shorter survival time was associated with self-reported sleep disturbance, neutrophil count, serum urea, creatinine, and C-reactive protein. Applying machine learning to actigraphy and sleep data combined with routine clinical data is a promising approach for the development of prognostic tools.

## 1. Introduction

Prognostication (i.e., estimation of survival) is an important aspect of the management of patients with cancer. It is of particular importance in advanced cancer where it has immediate implications for clinicians’ decisions about the treatment of the cancer, treatment of co-morbidities, so-called “ceilings of care”, and referral to palliative care services [1,2]. Furthermore, it has implications for patients (and families) in terms of current decision-making, advance care planning, and “getting one’s affairs in order”.

Healthcare professionals are inaccurate prognosticators, often overestimating survival [3], and the accuracy of estimates is inversely related to survival [2]. Healthcare professionals are relatively good at predicting if patients will die within a couple of days, but not so good at predicting if patients will live for a couple of months or longer.

Various prognostic tools/algorithms have been developed to improve prognostication in patients with cancer [2,4]: these tools vary in their content (e.g., objective items only; subjective items only; objective and subjective items). However, none of these tools have been shown to be consistently better than clinicians’ predictions of survival [2]. Current prognostication tools often include measures such as performance status, symptoms, venous blood sample data, and clinician-predicted survival [2,5]. The integration of other physiological and behavioural parameters, such as rest-activity rhythms (“diurnal or circadian”) and sleep parameters are yet to be considered in prognostic models. (The term ‘circadian’ is meant to refer to rhythms that persist in constant conditions. Rhythms assessed in the presence of environmental rhythms, as in the present study, are referred to as diurnal or 24 h rhythms, although increasingly these rhythms are also referred to as ‘circadian’)

Sleep-wake cycles and circadian rhythms have a key role in sustaining normal body function and homeostasis [6]. Deterioration of rest-activity rhythmicity (loss of rhythmicity) and fragmentation of the sleep-wake cycle may be a marker of deterioration of health and, indeed, a predictor of illness including cancer, as well as cancer survival [7,8,9].

Several studies in cancer patients have incorporated actigraphy to objectively assess daytime activity, 24 h variation in rest-activity, as well as nocturnal and daytime sleep [7]. A number of actigraphy-derived parameters have been used to quantify rest-activity rhythms in this population including acrophase (time of peak activity), amplitude (peak to nadir difference, i.e., height of activity rhythm peak), mesor (average activity over a 24 h period), and the “dichotomy index” (I < O). Of these parameters, the I < O is one of the most commonly studied rest-activity measures in cancer studies. The I < O has been identified as an independent prognostic biomarker for overall survival, particularly in patients with metastatic colorectal cancer [10,11]. The I < O is defined as the percentage of the activity counts measured when the patient is in bed that are inferior to the median of the activity counts measured when the patient is out of bed [12]. An I < O of ≤97.5% is indicative of a disrupted rest-activity circadian rhythm (i.e., increased fragmented sleep and reduced daytime activity patterns) [7]. However, the I < O has not been used to prognosticate per se, either alone or in combination with other items. Furthermore, few studies have explored the potential of actigraphy-derived sleep parameters as prognostic markers in advanced cancer patients [13].

The first aim of this study was to investigate the feasibility of using I < O and other actigraphy-derived parameters as stand-alone items, to prognosticate in patients with advanced cancer. The second aim of the study was to determine whether the I < O and other actigraphy and sleep parameters should be combined with established prognostic indicators, e.g., Eastern Cooperative Oncology Group performance status (ECOG-PS), modified version of the Glasgow Prognostic Score (mGPS), Prognosis in Palliative Care Study (PiPS) –B, as well as putative prognostic variables from routine clinical data derived from blood samples, to improve prognostic accuracy. To achieve this second aim we deployed regularised regression, a supervised machine learning approach which overcomes some of the limitations of classical multiple regression, to identify effective prognostic indicators and develop more robust prognostic algorithms [14].

## 2. Materials and Methods

### 2.1. Study Design and Setting

The study was a prospective observational study conducted in a medium-sized district general hospital/cancer centre in the United Kingdom. The study was sponsored by the Royal Surrey County Hospital and received ethical approval from the London–Bromley REC (reference number—16/LO/0243). The study was registered on the CancerTrials.gov registry (reference number—NCT03283683). The study was funded by the Palliative Care Research Fund (Prof. Davies—Royal Surrey County Hospital), including an unrestricted donation from the family of Mr. John Spencer.

### 2.2. Study Participants

Participants were recruited from outpatients at the study site. All patients that met the criteria for the study were eligible for entry into the study (convenience sampling, consecutive recruitment). The inclusion criteria were: (a) age ≥ 18 years; (b) diagnosis of locally advanced/metastatic cancer; (c) clinician estimated prognosis of more than 2 weeks but less than 1 year; and (d) known to a specialist palliative care team. The exclusion criteria were: (a) cognitive impairment; (b) physical disability that affected general activity; and (c) physical disability that affected non-dominant arm movement.

Patients were diagnosed with locally advanced/metastatic cancer according to NHS guidelines, which consider TNM staging. All patients who met the inclusion criteria were deemed eligible for entry into the study. Potentially eligible patients were identified by the clinical team and approached by a member of the research team and invited to participate in the study. Any patient referred to the specialist palliative care team was expected to die within the next twelve months (as per the General Medical Council definition for end-of-life care [14]).

### 2.3. Routine Data Collection

Written informed consent was obtained from participants prior to entry into the study. The initial review (day 0) involved a collection of routine clinical data: patient demographics, information about cancer diagnosis/treatment, information about co-morbidities/medication, assessment of Eastern Cooperative Oncology Group performance status (ECOG-PS) (by clinician and patient) [15], and completion of the Abbreviated Mental Test Score [16], the Memorial Symptom Assessment Scale—Short Form (MSAS-SF) [17], and the Global Health Status question from the PiPS-B algorithm [18]. The participant’s pulse was measured (as part of the PiPS), and a venous blood sample was taken to measure haemoglobin, white blood cell count (WBC), neutrophil count, lymphocyte count, platelet count, sodium, potassium, urea, creatinine, albumin, alanine aminotransferase (ALT), alkaline phosphatase (ALP), and C-reactive protein (CRP). The final review (day 8) involved further assessment of ECOG-PS (by clinician and patient), completion of the MSAS-SF, the Pittsburgh Sleep Quality Index (PSQI) [19], and a patient acceptability questionnaire. The blood test results were used to complete the PiPS-B scoring algorithm, and serum CRP and albumin were used to calculate the mGPS [20].

### 2.4. Wrist Actigraphy and Consensus Sleep Diary

Wrist actigraphy was used to measure physical activity and standard sleep measures. Participants were fitted with the Actiwatch Spectrum Plus^®^ (Philips Respironics, Bend, OR, USA) on the non-dominant arm after the initial review (day 0) and were instructed to wear the device for eight consecutive 24 h periods. The Actiwatch Spectrum Plus^®^ is a CE-marked device with an accelerometer (i.e., motion sensor) that samples movement at 32 Hz [21] with a sensitivity of 0.025 G (at 2 count level). Participants were also given a Consensus Sleep Dairy in order to provide confirmatory information about specific sleep parameters (e.g., number of awakenings, time of final awakening) [22]: the “diary” was completed for eight consecutive sleep periods. The Actiwatches were configured and data were retrieved using device-specific software (Actiware version 6.0.9: Philips Respironics, Bend, OR, USA). The Actiwatches were adjusted to provide an epoch length (sampling interval) of one minute, which is the most common epoch length used in studies of cancer patients [23]. The Consensus Sleep Diary was used in conjunction with the Actiwatch to assist in actigraphy data interpretation (i.e., determine the major sleep/wake periods) [24].

The data from the Actiwatches was downloaded into an Excel spreadsheet, and the following rest-activity parameters were calculated using a study specific SAS programme (SAS^®^ Version 9.4 Statistical Analysis Software, SAS Institute, Cary, NC, USA): I < O, r24 (an autocorrelation coefficient at 24 h, that is “a measure of the regularity and reproducibility of the activity pattern over a 24 h period from one day to the next”) [25], mean daily activity (MDA), and mean activity during daytime wakefulness. MDA was calculated as the average number of wrist movements per minute throughout the recording time [25], and the mean duration of activity during wakefulness was calculated as the mean activity score (counts/minute) during the time period between two major sleep period intervals [26]. In addition, the following sleep parameters were calculated both automatically from the Actiwatches (using the Actiware sleep scoring algorithm) and manually from the sleep diary [27]: bedtime (BT), get-up time (GUT), time in bed (TIB), sleep onset latency (SOL), total sleep time (TST), sleep efficiency (SE), wake after sleep onset (WASO), and number of awake episodes (NA). The sleep parameters derived manually solely from the sleep diary were: time tried to sleep, time of final awakening and terminal awakening (TWAK) [22]. See Table 1 for definitions of the sleep parameters.

**Table 1 cancers-15-00503-t001:** Definitions of actigraphy-derived sleep/consensus sleep diary parameters [22,26,28].

Sleep Parameter	Definition
Actigraphy and sleep diary	
Bed-time (BT) (hh:mm)	Clock time attempted to fall asleep based on actigraphy event marker or sleep diary
Get-up time (GUT) (hh:mm)	Clock time attempted to rise from bed for the final time based on actigraphy event marker or sleep diary
Time in bed (TIB) (hh:mm)	Duration between reported BT and GUT (reported in hours and minutes) or as self-reported in sleep diary
Sleep onset latency (SOL) (min)	Duration between reported BT and actigraph scored sleep onset time or as self-reported in sleep diary
Total sleep time (TST) (hh:mm)	Duration of sleep during the major sleep period calculated by Actiware;
	Sleep diary manual calculation: TIB minus (SOL plus WASO plus TWAK)
Sleep efficiency (SE) (%)	Proportion of time the patient is asleep out of the total time in bed (reported as a percentage) calculated by Actiware;
	Sleep diary manual calculation: TST divided by TIB × 100
Wake after sleep onset (WASO) (min)	Sum of wake times from sleep onset to the final awakening calculated by Actiware or as self-reported in sleep diary
Number of awake episodes (NA)	Number of continuous blocks of wake during the major sleep period calculated by Actiware or as self-reported in sleep diary
Sleep Diary	
Time tried to sleep (hh:mm)	Self-reported time participant began ‘trying’ to fall asleep
Time of final awakening (hh:mm)	Self-reported time participant last woke up in the morning
Terminal awakening (TWAK) (hh:mm)	GUT minus time of final awakening

### 2.5. Follow-Up

During the study period (from time of first patient recruited to six months after last patient recruited), participants’ survival status (and date of death, if applicable) was determined every three months by reviewing the hospital clinical records, and/or contacting the general practitioner.

## 3. Statistical Analyses

The sample size for the study (*n* = 50) was derived from guidance on sample sizes for feasibility studies (and represents the upper range) [29]. Statistical support was provided by statisticians, within the Research Design Service South-East (based in the Clinical Trials Unit at the University of Surrey). Descriptive statistics were used to explain much of the data (e.g., mean and standard error; median and range). The Intraclass Correlation Coefficient (ICC) was used to assess the robustness of I < O as a marker of the rest-activity rhythm, and its stability throughout the actigraphy recording. The Spearman’s Rank correlation coefficient was used to measure the association between I < O and other actigraphy-derived parameters. The Spearman’s rank correlation ‘*r*’ values were defined as follows: 0 ≤ *r* < 0.3 indicated a negligible correlation, 0.3 ≤ *r* < 0.5 a low correlation, 0.5 ≤ *r* < 0.7 a moderate correlation, 0.7 ≤ *r* < 0.9 a high correlation, and 0.9 ≤ *r* ≤ 1 a very high correlation [30]. Kaplan–Meier plots, a non-parametric statistical method, were used to estimate the probability of survival past a given time point along with the log rank test to compare the survival distribution of two groups. Statistical significance was evaluated at 5%.

The “per protocol set” refers to participants that wore the Actiwatch for the eight consecutive 24 h periods with the corresponding sleep diary, whilst the “full analysis set” refers to participants that wore the Actiwatch for at least three consecutive 24 h periods (i.e., 72 h) and completed the corresponding sleep diary for the actigraphy rest-activity and sleep analysis, or for at least three consecutive or non-consecutive nights in the sleep diary for the subjective sleep analysis (i.e., calculation of the sleep diary parameters).

## 4. Machine Learning Methods and Data Analysis

Cox regression has been the standard approach to survival analysis in oncology. However, Cox regression has a number of limitations. In particular, it is not an adequate approach for situations in which the number of predictors is high relative to the number of observations, as is the case in this feasibility study. We therefore opted to use simple alternative methods that can (1) adequately deal with situations in which the number of predictors is large relative to the number of observations and (2) yield models that are interpretable, i.e., are not ‘black box models’. Penalised (Regularised) regression models represent such an approach.

A supervised machine learning algorithm was used to develop a predictive model, where the collated subjective and objective parameters (i.e., routine clinical data and actigraphy-derived rest-activity and sleep parameters) were individual predictor variables and survival was the ‘response’ variable [31]. Sixty-six predictor variables were tested for potential predictive value (Appendix A, see Table A1 for descriptive statistics of the numerical predictor variables). Overall survival was defined as the time from initial review (day 0) to death or until 14 May 2020 for patients that remained alive until the end of the study.

### 4.1. Machine Learning Dataset

All patients recruited into the study (*n* = 50) were used for the machine learning analysis. The predictor variables were classified into the relevant variable type (e.g., binary, categorical_nominal, etc.) and entered into a .csvfile in Excel. Binary variables, such as ‘use of opioid analgesia’ were transformed into dummy variables (0 or 1). Categorical_ordinal variables with a numerical ranking, such as ECOG-PS were labelled using the ‘LabelEncoder’ approach, where the output integer value from the LabelEncoder function was used to reflect the ordering of the original integer. Categorical_ordinal variables with non-numerical values, such as PSQI sleep disturbance, were assigned a numerical ranking. Numerical_continuous variables involving sleep/wake times were entered in the 24 h format. Missing data values were imputed with the average of the group or with the corresponding subjective/objective data from the same participant. Missing data accounted for <4% of the dataset.

### 4.2. Regularised Regression Methods

Regularised regression was used to reduce “overfitting” and aid the generalisability of the model. ‘Regularisation’ corresponds to a penalty that limits the overall weight that can be assigned across all predictor variables in the model, which reduces model complexity (compared to traditional multivariate regression). For some regularised regression approaches, the penalty can drive the weight of a variable to zero, effectively selecting the optimal combination of predictor variables that can be used to predict the given outcome.

Here, three regularised multivariate regression methods were applied and compared: ridge regression, least absolute shrinkage and selection operator (Lasso) and elastic net. The ridge regression algorithm includes all the predictor variables, shrinking the coefficients towards (but not set at) zero in a continuous manner [32]. The Lasso-derived algorithm combines the method of shrinkage with the sub-selection of predictor variables, using a penalty ‘*L*_1_ norm’ [32,33], creating a ‘sparse’ model (i.e., selecting only a few variables from the dataset) [32]. The elastic net algorithm is broadly a combination of the ridge and Lasso [34]. This method simultaneously performs continuous shrinkage and feature selection, selecting groups of correlated variables, using a penalty of ‘*L*_1_ norm’ and ‘*L*_2_ norm’ [34]. Highly correlated predictor variables are averaged and entered into the model to remove any deviances caused by extreme correlations [35]. Since survival data are censored, i.e., at the end of the observation period some participants may still be alive, we applied regularised Cox regression using the glmnet package in R.

### 4.3. Model Development

The models were validated using a *k*-fold (10 folds used) cross-validation approach [32]. For each of the 50 individuals, the predicted survival was based on a model which was constructed on ‘*k* − 1′ subjects, i.e., the model was blind to the participant and the participant did not contribute to the estimation of the prediction. All analyses were carried out within the statistical computing environment R (version 3.6.2). For machine learning, ridge, Lasso and elastic net (alpha = 0.5) regression the package glmnet (version 2.0) was used. Here, an exhaustive search for lambda able to produce the minimum Mean Cross-Validated Error (CVM) was performed. All subjects were used as the training set to build a final model, then *k*-fold cross-validation for performances (CVM) was performed. Analyses were performed with different settings of elastic net mixing parameter (alpha), which were elastic net (alpha = 0.5), Lasso (alpha = 0.99) and ridge (alpha = 0.01). The models generated a predicted hazard, which was compared to the actual survival in days using Pearson’s correlation coefficient. To estimate the intra-variable variation in their contribution to the predictor, we computed the mean cross-validated error of the weights of each of the variables that were consistently identified in all 50 participants.

## 5. Results

A total of 50 patients were recruited to the study, and 49 participants completed the study (Figure 1): the full analysis set consisted of 44 participants, whilst the per protocol set consisted of 37 participants. See Table 2 for characteristics of the participants. A total of 46 participants were followed up for 12 months (40 in the full analysis set, 33 in the per protocol set), and 34 died within this time period (28 in the full analysis set, 22 in the per protocol set). Unless otherwise stated, the following results relate to the full analysis set.

### 5.1. Acceptability of Actigraphy and Sleep Diary Acceptability

Actigraphy data were missing from one participant due to a technical problem. Forty-two (84%) participants reported that the Actiwatch was “comfortable to wear”, and only four (8%) reported that the Actiwatch interfered with their normal activities. No adverse effects were reported from using the Actiwatch. Fourteen (28%) participants reported that the Consensus Sleep Diary was difficult to complete, and two (4%) subjects reported that the diary interfered with their normal activities.

### 5.2. Univariate Analyses of Actigraphy Parameters

#### 5.2.1. Characteristics of the Dichotomy Index (I < O) and Correlation with Other Actigraphy and Sleep Parameters

Table 3 shows the results for the I < O. Forty-two (95%) participants had an I < O of ≤97.5%, indicating a disrupted rest-activity circadian rhythm [7]. The I < O can be considered a stable variable since the intraclass correlation coefficient for values obtained over eight days using the per protocol set, was 0.93 (95% CI: 0.88–1.00; *p* < 0.0005), which is considered an “excellent” correlation [36]. In fact, there was a “high” positive correlation between the I < O for the first three days (72 h) and for the full eight days (Spearman’s correlation: *r* = 0.82; *p* < 0.0005) [31]. Moreover, there was a “high” positive correlation between the I < O on weekdays and on the weekend (Spearman’s correlation: *r* = 0.76; *p* < 0.0005). Additionally, there was a “very high” positive correlation between the I < O calculated using 24 h of data, and the I < O calculated using 20 h of data, i.e., excluding the one-hour periods before/after going to bed, and the one-hour periods before/after getting out of bed (Spearman’s correlation: *r* = 0.98; *p* < 0.0005).

There was a “moderate” positive correlation between the I < O and the r24 (Spearman’s correlation: *r* = 0.66; *p* < 0.0005), and the mean activity during wakefulness (Spearman’s correlation: *r* = 0.51; *p* < 0.0005). However, there was only a “low” positive correlation between the I < O and the mean daily activity (Spearman’s correlation: *r* = 0.43; *p* = 0.003). Other standard actigraphy parameters correlated with the I < O were SE, i.e., number of minutes of sleep divided by total number of minutes in bed (Spearman’s correlation: *r* = 0.47, “low” correlation; *p* = 0.001), and WASO, i.e., number of minutes awake after sleep onset during sleep period (Spearman’s correlation: *r* = −0.51, “moderate” correlation; *p* < 0.0005).

#### 5.2.2. I < O: Predictor of Survival and Correlation with ECOG-PS

Amongst participants that completed one year of follow-up (*n* = 40), there was no significant difference in overall survival between those separated into two groups (based on the median I < O; log rank test, *p* = 0.917), or four groups (based on the quartiles of the I < O; log rank test, *p* = 0.838). However, I < O had a “moderate” negative correlation with the physician assessed ECOG-PS (Spearman rank correlation: *r* = −0.63; *p* < 0.0005). The ECOG-PS was an independent prognostic indicator in this cohort of patients (log rank test, *p* < 0.0005). The median survival for participants with an ECOG-PS of 1 (end of study) was 141 days, ECOG-PS of 2 was 135 days, ECOG-PS of 3 was 57 days, and ECOG-PS of 4 was 17 days.

#### 5.2.3. Autocorrelation Coefficient at 24 h (r24)

The median r24 was 0.16 (range 0.04–0.37). Amongst participants that completed one year of follow-up (*n* = 40), there was no significant difference in overall survival between those separated into two groups (based on the median r24; log rank test, *p* = 0.318), or four groups (based on the quartiles of the r24; log rank test, *p* = 0.800).

#### 5.2.4. Other Actigraphy Parameters

None of the other actigraphy-derived sleep parameters were associated with a decreased overall survival: (a) TIB (log rank, *p* = 0.574: based on group median of 9 h 29 min); (b) TST (log rank, *p* = 0.147: based on normative cut-off value of ≥6.5 h [28]; (c) SOL (log rank, *p* = 0.283: based on normative cut-off value of ≤30 min [28]; (d) SE log rank, *p* = 0.224: based on normative cut-off value of ≥85% [28]; (e) WASO (log rank, *p* = 0.549: based on normative cut-off value of >30 min [28]; and (f) NA (log rank, *p* = 0.972: based on group median of 23 episodes).

### 5.3. Multivariate Predictors of Survival: Machine Learning Results

In the machine learning dataset, 46 participants had died within the specified time period of follow-up (i.e., by 14 May 2020). The Lasso model selected 22 predictor variables, with 14 variables consistently selected in all 50 participants during the process of validation (Figure 2). These involved eight predictor variables associated with greater survival time and six predictor variables, associated with a reduced survival time. The predictor variables associated with increased survival time, i.e., smaller hazard (in order of the coefficient associated with the predictor variable) were: later sleep diary time of final awakening, later actigraphy get up time, longer PiPS-B clinician’s estimate of survival, better PSQI subjective sleep quality, greater PiPS-B global health status score (indicating better health), better actigraphy sleep efficiency, and higher haemoglobin values. The variables associated with reduced survival time were more frequent PSQI sleep disturbance wake middle of the night/early morning, higher neutrophil count, higher serum urea, serum creatinine, and serum C-reactive protein. On the contrary, a larger MSAS-SF total symptom distress was associated with a lower risk of death and a higher I < O was associated with a worse prognosis. The predicted median hazard was 0.00052, and the model was able to successfully differentiate between participants with a shorter/longer overall survival (log rank *p* < 0.0001) (Figure 3). Figure A1 shows the correlation between the actual survival and predicted hazard (Pearson’s correlation coefficient *r* = −0.5; *p* = 0.0002).

The ridge model consistently identified 28 predictor variables in all 50 participants (Figure 4). During the process of validation, the top 10 variables consistently selected involved seven predictors associated with longer survival time and three predictors associated with shorter survival time. The seven predictor variables associated with longer survival time (in order of the coefficient associated with the predictor variable) were: actigraphy get-up time, sleep diary time of final awakening, sleep diary get-up time and PSQI usual get-up time; PiPS-B clinician’s estimate of survival, PSQI subjective sleep quality and PiPS-B global health status score. The 3 predictor variables associated with shorter survival time were: use of opioid analgesia, modified Glasgow Prognostic Score and physician-assessed ECOG-PS day 8. The predicted median hazard was 0.44; however, there was no significant difference in overall survival when a median split was applied (log rank, *p* = 0.0914) (Figure 5). Figure A2 shows the correlation between the actual survival and predicted hazard (Pearson’s correlation coefficient *r* = −0.5; *p* = 0.0002).

The elastic net model selected 10 predictor variables, with 6 variables being consistently selected during the process of validation: the two consistently selected predictor variables associated with longer survival time were (in order of the coefficient associated with the predictor variable): later actigraphy get-up time and greater PiPS-B global health status score; the 4 consistently selected predictor variables associated with shorter survival time were: higher serum urea, neutrophil count, serum C-reactive protein, and serum creatinine (Figure A3). The predicted median hazard was 0.408, but there was no significant difference in overall survival (log rank, *p* = 0.9877) (Figure A4). Figure A5 shows the correlation between the actual survival and predicted hazard (Pearson’s correlation coefficient *r* = −0.08; *p* = 0.5808).

## 6. Discussion

The results of this study show that univariate approaches to survival prediction, based on, for example, the I < O, are not very powerful; whereas, multivariate approaches appear to hold promise. To the best of our knowledge, this is the first study describing the application of supervised machine learning methods, involving a combination of actigraphy-derived rest-activity and sleep parameters, and data collected in routine clinical practice (i.e., simple questionnaires such as the MSAS-SF, ECOG-PS, PSQI, venous blood sampling) to prognosticate patients with advanced cancer, receiving supportive and palliative care [37]. Our study confirmed certain established novel predictors and identified some for survival in this group of patients and points to the importance of sleep characteristics for prognostication. The results of the study also confirm that clinicians are inaccurate prognosticators [3], since 11 (24%) participants were still alive at 1 year (despite the inclusion criteria of clinician estimated prognosis of more than 2 weeks but less than 1 year).

The literature had suggested that actigraphy-derived parameters, and the I < O index in particular, could be used as predictors because a low I < O is associated with increased morbidity (worse symptoms, worse quality of life), and with decreased survival [7]. At the outset of this study, we therefore focused on the I < O and other parameters describing the robustness of the rest-activity. We indeed observed a very high prevalence (i.e., 95%) of disrupted rest-activity rhythms in these advanced cancer patients, which is much higher than the reported prevalence of 19.1–54.9% [7]. This disparity undoubtedly reflects different populations, with our population having more advanced disease (and worse performance status) than previous studies [11,38]. However, in the univariate analyses of the data in our study there was no direct association between I < O and survival. Furthermore, other actigraphy-derived parameters, when used in isolation, are also not very accurate in the population.

However, the results of the study suggest that novel models developed through machine learning can facilitate improvements in prognostication. Penalised regression methods implement a feature selection strategy, providing a combination of subjective and objective predictor variables of survival that are ranked based on their contribution to the model. The models manage collinearity within the dataset, which is particularly useful in datasets involving terminal cancer patients, where often the number of features exceeds the relative sample size. The best performing method was Lasso regression which reduces the coefficients of variables with a minor contribution to zero and thereby creates a simple ‘model’ with only a few variables. Sleep parameters were amongst the most important variables, not only in the Lasso model but also in the more complex elastic net and ridge models. These measures primarily represented positive predictors of survival. Sleep diary final awakening (lasso and ridge) and actigraphy-derived GUT (all models) were found to have particular prognostic relevance in our study, suggesting that a later sleep diary determined ‘time of final awakening’ and a later actigraphy-derived ‘get-up time’ are associated with a lower risk of death and improved survival. Furthermore, actigraphy-derived SE, which may be considered an objective measure of sleep quality, was selected as a positive predictor of survival in the lasso model (i.e., greater sleep efficiency was associated with enhanced survival) for our population. Whilst actigraphy-derived sleep quality, as opposed to sleep quantity, has been reported to have prognostic significance in advanced breast cancer patients [13], we identified quantitative sleep measures as important contributors to survival prediction.

Studies have reported actigraphy-derived circadian disruption [10,12,39] and fragmented sleep [13] to have prognostic implications in cancer patients, yet little is known about the prognostic impact of subjective sleep measures. A recent study identified the PSQI sleep duration component as a prognostic indicator in a cohort of advanced hepatobiliary/pancreatic cancer patients [40], yet a novel finding in our study was the selection of other sleep parameters from the PSQI: (1) usual get up time and (2) subjective sleep quality, where a later get up time and very good sleep quality are associated with longer overall survival, and (3) PSQI sleep disturbance components—pain, cannot breathe comfortably and wake up in the middle of the night or early morning—were associated with poorer survival. Furthermore, subjective sleep parameters, as opposed to actigraphy-derived sleep parameters, were more commonly identified in all participants in the ridge model.

Venous blood sample measurements were also significant contributors to predicting survival in our study. Previous studies have reported moderate evidence for the prognostic significance of an elevated C-reactive protein (CRP) and leucocytosis being associated with a shorter survival [1,5,18,41]. Whilst our study was able to echo these findings, we were able to further identify novel biomarkers, such as an elevated urea and serum creatinine, that may also be associated with a poorer survival, and raised haemoglobin that may be associated with a lower risk of death. Blood sampling is generally deemed ‘inappropriate’ when patients are in their last days/weeks of life [42], regardless only one of the 94 patients screened for our study, declined participation. Our findings endorse further evaluation of biological parameters from venous blood sample data, as they may be beneficial to improving prognostication in these patients.

Although our multivariate findings controversially imply that a higher I < O is associated with a shorter predicted survival time, all participants in our population had poor health, i.e., an I < O of <99%, which has recently been identified as an optimal cut-off for distinguishing between healthy controls and patients with advanced cancer [43]. Further inspection of our data identified that all our participants, whether they had shorter or longer survival had disrupted rest-activity rhythms, equally both groups had moderate symptom distress as measured by TMSAS, inevitably expected in an advanced cancer population. Therefore, whilst it may be a simple way of quantifying rest-activity rhythms, I < O may be a more meaningful prognostic indicator during the earlier trajectories of cancer, as opposed to the progressive stages.

In summary, our data suggests that subjective sleep parameters, measured using the consensus sleep diary and the PSQI, and actigraphy-derived sleep parameters may be especially useful when combined with routine clinical data using machine learning approaches, with no substantial additional costs or burden to the health service. Thus, further investigation of these parameters as prognostic indicators is warranted. Indeed, we plan to undertake a larger (definitive) study in the near future. Sleep-wake disturbances and circadian dysregulation are deemed to have a reciprocal relationship [43,44] and our findings are suggestive of sleep/circadian rhythm parameters as potential prognostic indicators. Whether improving the patient’s sleep disturbance may improve overall survival remains an open question. Rehabilitation of the circadian system by means of behavioural and pharmacological strategies, to re-synchronise the circadian system, may ultimately improve circadian function and sleep, as well as overall survival [44,45].

The Lasso model was the only model able to successfully differentiate between long and short survival in our study, and the correlation between observed and predicted hazard was only significant for the Lasso and ridge models. The Lasso model is ‘sparse’ (i.e., only a few variables from the dataset are selected) [32] and therefore may be favourable if a consolidated model were needed to aid prognostication. However, the Lasso selects at most ‘*n*’ variables before it saturates; therefore, the number of predictors is restricted by the number of observations [32]. The ridge model, therefore, may be beneficial due to the greater inclusivity of variables, at the expense of an increased risk of overfitting. Indeed, the absence of significant results cannot be overlooked with the small sample size in this feasibility study. In the definitive study, all three supervised machine learning methods would be deployed after the recruitment of a larger sample size as well as the inclusion of additional variables that may be clinically relevant (e.g., stage of disease, number of comorbidities, nutritional status, presence of specific symptoms/problems) [1,2], More data would enable robustness of the predictive ability of the models to be assessed as well as enable generalisability of our findings with further confidence in our observations.

Interestingly, a recent systematic review described the prediction of survival to be a process as opposed to an event, and that predictors of survival may develop as the disease progresses [5]. Therefore, there may be added value in predicting the trajectory of death, as opposed to the time of death in future studies. Machine learning approaches would be particularly valuable in such cases, where relevant predictor variables may be identified as the disease trajectory evolves only to ultimately enhance our true understanding of prognostication.

A few limitations need consideration. Firstly, our small sample size is unlikely to capture the true variance of the population. Secondly, the Lasso and elastic net models involve only a subset of predictors and the value of the coefficient associated with each of these predictor variables is dependent on the presence of the other (non-zero) predictor variables in the model. Our results are essentially correlational and demonstrate that the relevant predictor variables (above non-zero coefficient value) may be associated in a positive or negative way with the risk of death. Thirdly, imputation of missing data values with the sample population average may not have been a true reflection of the individual sample’s actual score nor using subjective data to impute objective values, particularly if the tools were measuring different timeframes, i.e., actigraphy (over a one-week duration) versus the PSQI questionnaire (measures on average over the previous one month). The *K*-fold cross-validation approach also has some limitations. As it is executed ‘*k*’ times (where ‘*k*’ is the number of subsets of observations), this approach may not be resourceful in a small dataset. Furthermore, *K*-fold cross-validation is likely to have a high variance as well as a higher bias, given the small size of the training set from a small dataset. Therefore, the number ‘*k*’ highly influences the estimation of the prediction error, and the presence of outliers can lead to a higher variation. Indeed, it can be a challenge to find the appropriate ‘*k*’ number to reach a good ‘bias-variance’ trade-off. In future studies, it will be essential to include an independent validation set.

## 7. Conclusions

This study suggests that subjective sleep parameters, measured using the consensus sleep diary and the PSQI, and actigraphy-derived sleep parameters may be useful for prognostication in patients with advanced cancer, and that it may be especially useful when combined with routine clinical data and machine learning approaches.

## Figures and Tables

**Figure 1 cancers-15-00503-f001:**
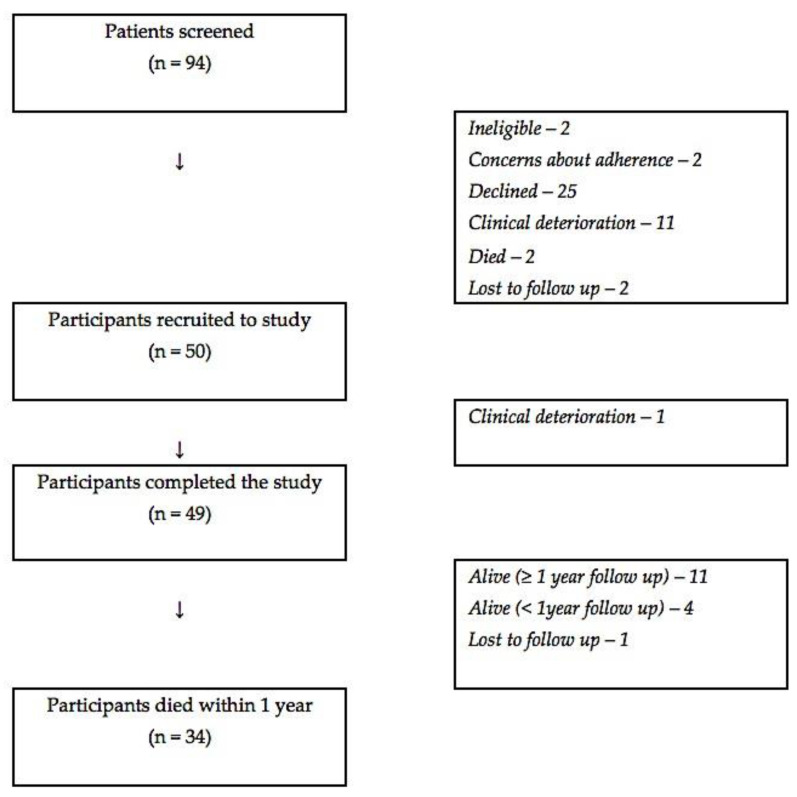
Study flow chart.

**Figure 2 cancers-15-00503-f002:**
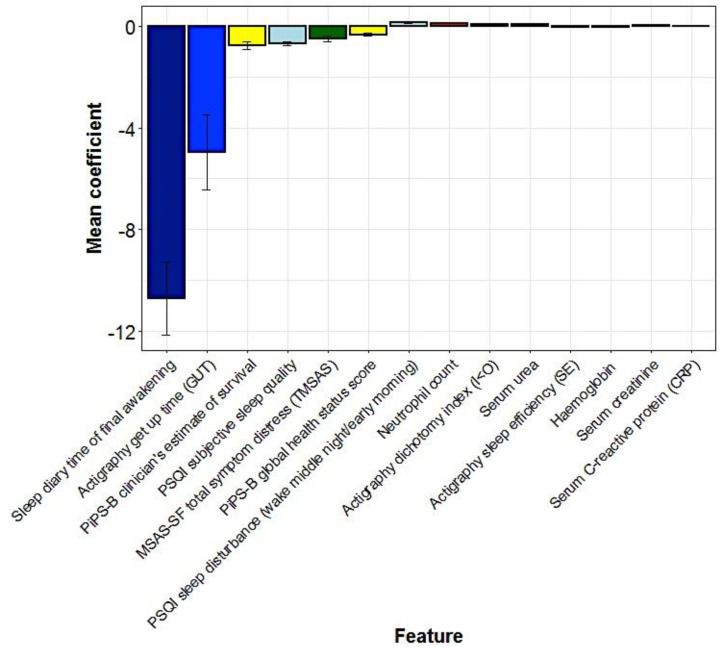
The mean cross—validated error (CVM) of predictor variables for hazard selected by the Lasso model.

**Figure 3 cancers-15-00503-f003:**
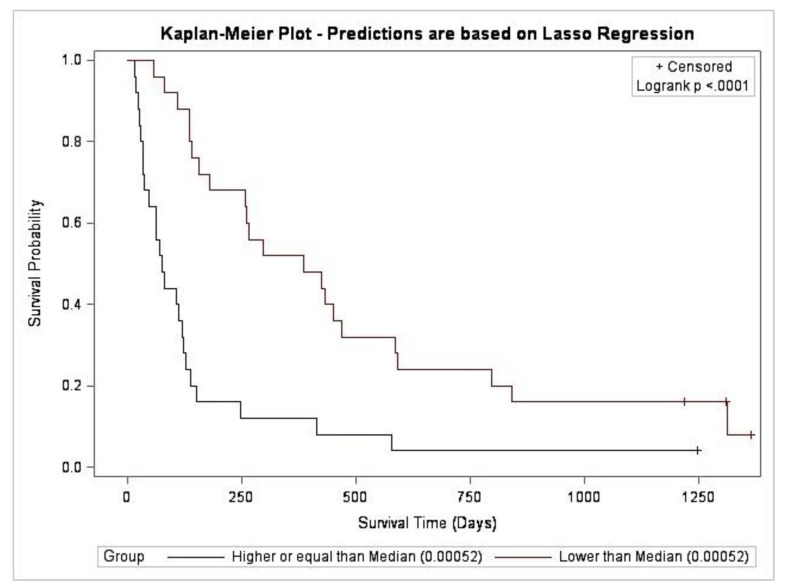
Kaplan–Meier curve comparing survival probability predicted by the Lasso-derived algorithm (log rank, *p* < 0.0001).

**Figure 4 cancers-15-00503-f004:**
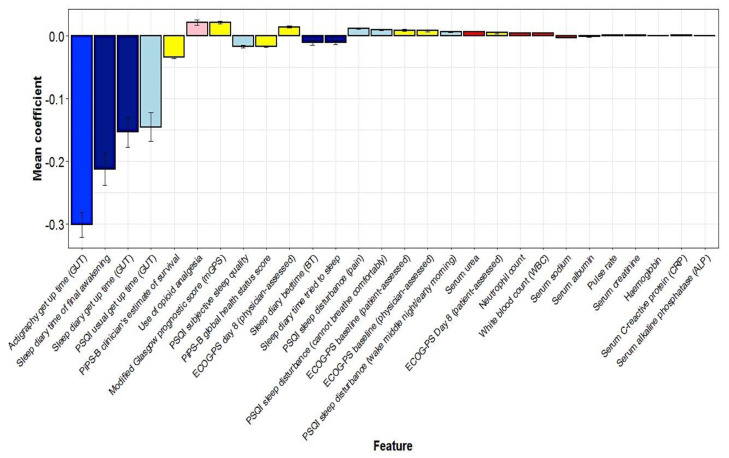
The mean cross—validated error (CVM) of predictor variables for hazard selected by the ridge model.

**Figure 5 cancers-15-00503-f005:**
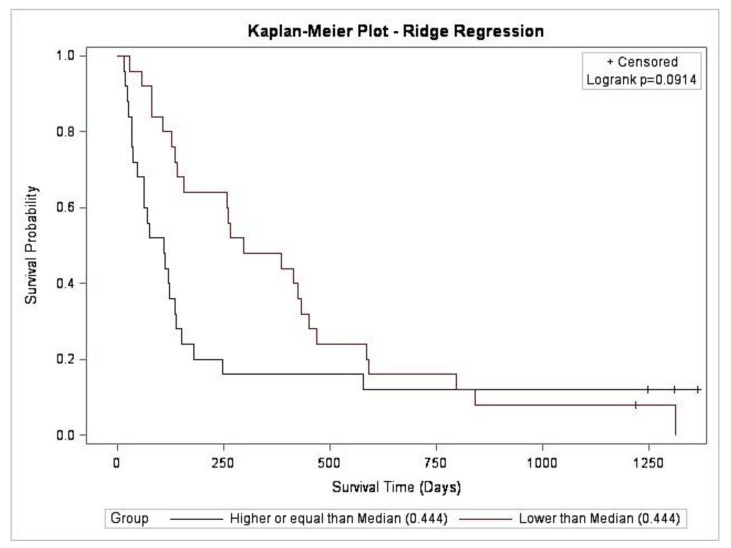
Kaplan–Meier curve comparing survival probability predicted by the ridge-derived algorithm (log rank, *p* = 0.0914).

**Table 2 cancers-15-00503-t002:** Participant characteristics.

Characteristic	All Participants (*n* = 50)	“Full Analysis Set” (*n* = 40)
Age	Median—63 yr	Median—66 yr
	(range 40–81 yr)	(range 43–81 yr)
Sex	Female—21 (42%)	Female–17 (39%)
Male—29 (58%)	Male—27 (61%)	
Cancer diagnosis	Breast—6 (12%)	Breast—6 (14%)
	Endocrine—1 (2%)	Endocrine—1 (2%)
	Gastrointestinal—16 (32%)	Gastrointestinal—14 (32%)
	Gynaecological—6 (12%)	Gynaecological—4 (9%)
	Haematological—2 (4%)	Haematological—2 (5%)
	Head and Neck—3 (6%)	Head and Neck—2 (5%)
	Lung—6 (12%)	Lung—6 (14%)
	Skin—2 (4%)	Skin—2 (5%)
	Urological—8 (16%)	Urological—7 (16%)
ECOG-PS	0–0 (0%)	0–0 (0%)
(Physician-assessed	1–26 (52%)	1–24 (55%)
at baseline)	2–13 (26%)	2–10 (23%)
	3–11 (22%)	3–10 (23%)
	4–0 (0%)	4–0 (0%)

Note: Percentages may not sum to 100 due to rounding.

**Table 3 cancers-15-00503-t003:** Dichotomy Index (I < O) data.

I < O Parameter	Full Analysis Set (*n* = 44)	Per Protocol Set (*n* = 37)
Mean	88.90%	89.90%
(+/− standard error)	(+/− 1.04)	(+/− 0.97)
Minimum	70.90%	70.90%
25th Centile	86.90%	87.40%
Median	90.40%	90.80%
75th Centile	93.60%	93.60%
Maximum	98.10%	97.60%
Distribution	Non-normal	Non-normal
	(Shapiro-Wilk	(Shapiro-Wilk
	test: *p* = 0.001)	test: *p* = 0.001)

## Data Availability

The data presented in this study are available on request from the corresponding author. The data are not publicly available due to the nature of the consent obtained from participants.

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
