# Peer review of "Prognostication in Advanced Cancer by Combining Actigraphy-Derived Rest-Activity and Sleep Parameters with Routine Clinical Data: An Exploratory Machine Learning Study"

_cancers, 2023, doi:10.3390/cancers15020503_

Round 1

Reviewer 1 Report

The authors report the impact of activity-derived rest-activity in cancer prognostication. More specifically, using actigraphy-derived measures such as the dichotomy index, and other sleep parameters combined with routine clinical data and machine learning approaches to develop new prognostic tools.

Global comment:

The manuscript is well written and easy to understand. The authors have done a lot of work to establish the state of the art. The description of the applied methodology is also very well documented. The authors argue clearly and precisely about the use of machine learning techniques and how they define their model. However, some points need clarification to understand the analysis.

Prediction model

In the results part, the authors present the correlation between actual survival and predicted survival based on the ridge model. It would be interesting to add additional information to evaluate this prediction model, such as a table grouping the sensitivity, specificity, and Matthews's correlation for example.

Author Response

Comments and Suggestions for Authors

The authors report the impact of activity-derived rest-activity in cancer prognostication. More specifically, using actigraphy-derived measures such as the dichotomy index, and other sleep parameters combined with routine clinical data and machine learning approaches to develop new prognostic tools.

Global comment:

The manuscript is well written and easy to understand. The authors have done a lot of work to establish the state of the art. The description of the applied methodology is also very well documented. The authors argue clearly and precisely about the use of machine learning techniques and how they define their model. However, some points need clarification to understand the analysis.

Authors’ Response:

We thank the reviewer for the positive comments.

Prediction model

In the results part, the authors present the correlation between actual survival and predicted survival based on the ridge model. It would be interesting to add additional information to evaluate this prediction model, such as a table grouping the sensitivity, specificity, and Matthews's correlation for example.

Authors’ Response:

Sensitivity, specificity, and Matthews’ correlation are metrics for categorical data. Here the variables are numerical, i.e., how many days will someone survive. If one wanted to provide information above and beyond a correlation one could provide a Bland-Altman type of analysis and present bias, etc. but we feel that for a ‘feasibility/explorative study, the information as provided is sufficient to illustrate the potential of the approach. 

Reviewer 2 Report

I think the report of this study is very good, and it is a very meaningful topic. The article describes the research methods and conclusions reasonably.

Some minor problems: 1) In figure 1, there are some problems with the interface layout; 2) In terms of the choice of machine learning methods, why the choice description in this article is not clear enough, whether it can show that the machine learning method is more effective than the classic COX, and whether it is verified; 3) This article mentions the problem of limitations. It is true that there are very few samples. Is it necessary to adopt the method of machine learning? Then the problem of insufficient sample, whether it can be optimized through the algorithm of this paper, the text needs to reflect

Author Response

Comments and Suggestions for Authors

I think the report of this study is very good, and it is a very meaningful topic. The article describes the research methods and conclusions reasonably.

Authors’ response:

We thank the reviewer for the positive comments.

Some minor problems:

1) In figure 1, there are some problems with the interface layout;

Authors’ response:

Thank you for pointing this out. We have corrected this problem in the new version of the manuscript.

2) In terms of the choice of machine learning methods, why the choice description in this article is not clear enough, whether it can show that the machine learning method is more effective than the classic COX, and whether it is verified;

Authors’ response:

We agree with the reviewer that the Cox regression has been the standard approach to survival analysis in oncology. However, Cox regression has a number of limitations. In particular, it is not an adequate approach for situations in which the number of predictors is high relative to the number of observations, as is the case in this feasibility study.  We therefore opted to use simple alternative methods that can 1) adequately deal with situations in which the number of predictors is large relative to the number of observations and 2) yields model that are interpretable, i.e., are not ‘black box models’ (cf PMID: 33772109). Penalised regression models represent such an approach. We have now added a sentence to the Methods to explain this.

3) This article mentions the problem of limitations. It is true that there are very few samples. Is it necessary to adopt the method of machine learning? Then the problem of insufficient sample, whether it can be optimized through the algorithm of this paper, the text needs to reflect

Authors’ response:

We agree with the reviewer that in this study there are not many observations. However, this is a feasibility study, and a characteristic of a feasibility study is that there are not many samples. We wanted to demonstrate the feasibility to conduct this type of study and the feasibility of constructing a ’multivariate predictor’ for survival. The method we used is regularised or penalised regression.  This simple machine learning method is specifically designed to deal with situations in which there are many potential predictors (features) and relatively few observations (c.f. PMID: 33216811) as is the case in this feasibility study. In the revised version of the manuscript, we have explained this in more detail. 

Reviewer 3 Report

In this work, the authors combine data from different sources (actigraphy, sleep journal, clinical data) to predict patient survival in advanced stage cancer patients. This type of modelling could add value to knowledge in the field, since it could potentially provide evidence advocating that a more complete monitoring of a patient could improve his/her outcome.

Unfortunately, there are a few issues that refrain me from suggesting this article for publication. There are two major flaws in the methodology of the proposed analyses. These are explained as follows, together with some other observations. 

Major comments

·         L97: From reading the introduction, it is not clear to me why the authors decided to focus on patients with advanced cancer.

·         L114: The authors should provide a complete description of the data here, similar to Table 1, explicitly including the descriptive statistics of each feature (e.g., mean and SD for numerical normal features or, probably more realistic, median and IQR for non-normal features; for categorical features the proportion of patients across different groups; the percentage of missing values in each feature). The description of the data used should always go in the M&Ms and never in the Results, since this doesn’t answer any research question, it just provides the reader with information about the used data.

·         L118: Moreover, how were patients with locally advanced/metastatic cancer identified? I suppose this was done based on their TNM staging, but this isn’t explicitly mentioned in the text

·         L124: It is not clear how many time points were involved in the data collection. Were they only 2 (day 0 and day 8)? If so, why at those days? Were the data of all patients collected in time?

·         L159: How were the mean daily activity, and mean activity during daytime wakefulness calculated?

·         L160: It is not clear which parameters were calculated automatically from the software and which were calculated manually from the sleep diary. These distinction should be made clear. Moreover, it should be clearly stated how parameters such as sleep latency or sleep efficiency were calculated – or at least how are they defined.

·         Materials and Methods: the organization of this section is a bit messy. Section 2.4 should be a subsection of 2.3. The authors make a distinction between Statistical and Machine Learning methods. This is a gray area and the definition is not crystal clear. Actually, some people might argue that regressions are not part of ML, but more of classical statistics. To avoid this discussion, I suggest that the description of this methods are just grouped under a subsection such as “Data analysis” or so. More importantly, either statistical or ML, this is one of the most important parts of the M&Ms and they should be incorporated into this section as such (and not being reported in a different section)

·         L182: It is not clear to me why did the authors compare survival probability between two groups corresponding to different time points.

·         However, my two biggest issues with the current analysis are the following:

o   The dataset consisted of only 50 patients. This is a very small number for the type of analysis being done. This is particularly worrisome when using models with such a large number of features (66 in one case), which leads to the curse of dimensionality, making it very hard to draw solid conclusions from such a model. In a best case scenario, adding a large amount of patients would solve this issue. However, I know that this is probably very impractical. Thus, I would suggest that the authors drastically reduce the number of features used in their models. They could either pick a few features that have the largest predictive power based on the univariate analysis or they could also engineer features that combine information from some of them.

o   My second largest issue is with the choice of model. The regression model used in this study is inappropriate, since it does not take into account censorship, a crucial issue when doing survival analysis. The authors should have used a Cox Proportional Hazards regression at the very least. If they are interested in the ML aspect, they could further explore the use of Random Survival Forests, Survival Support Vector Machines, or Extreme Gradient Boosting with Cox, to mention a few.

·         Given these two important issues, I am afraid that I cannot judge the validity of the drawn conclusions, since they come from a faulty analysis pipeline. Therefore I cannot review the remaining of the manuscript. I encourage the authors to redo their analysis in a proper way and resubmit their manuscript accordingly. I would be happy to review it then.

·         Lastly, I would suggest that if the authors want to compare the performance of different models, they should look into a model-agnostic explainability layer, such as SHAP. This could provide better insights into feature importance for the models, as well as a knowledge representation of what the models do to reach their predictions.

Minor comments

·         All through the manuscript:

o   Replace “prognostication” with “prognosis”, it is a more suitable word

o   Please us the Oxford comma for lists

o   The authors often use very long sentences, which are hard to read. I have pointed a few in my comments, but not all of them. Please make sure that overall sentences are shortened.

o   E.g., and i.e., should always have a comma at the end

o   No need for spaces before/after a slash

o   Abbreviations were often defined twice in the text. They should only be defined the first time they are used, after that, the only the abbreviated version should be used.

o   When reporting physical quantities, there should always be a space between the number and the unit

·         L24: This sentences is too long

·         L30: This sentence could be written better. E.g., “Our findings confirmed the importance of certain well-established survival predictors, as well as identified new ones, in this patient subpopulation”

·         L36: Provide one or two reasons why they remain elusive, since it is not clear how adding actigraphy data would help this issue.

·         L37: “parameters” repeated in the same sentence

·         Keywords should be in alphabetical order. Moreover, they should be different from words used in the title. Otherwise, there is no point.

·         L54: Prognostication is a much broader term and cannot be used as a substitute for survival estimation. If the authors wish to focus or estimating survival, they should refer as such and avoid the more generic term of prognostication (or prognosis)

·         L55: “and indeed all patients with life-limiting conditions” should be removed. It is obvious and the focus of the paper is cancer patients.

·         L60: This paragraph is very convoluted. It could be easily understood with just the last sentence. Moreover, “that” --> “if”

·         L66: It is not clear what the authors mean with objective and subjective items

·         L69: This sentence is too long

·         L71: The fact that activity and sleep parameters aren’t included in current models should be mentioned after the explanation of why are they important

·         L72: Footnotes with numbers should be favored

·         L77: survival also in cancer --> cancer survival

·         L81: Diurnal or circadian was already defined before

·         L86: The 1 points to no footnote

·         L91: The use of “however” should be favored at the beginning of the sentence

·         L92: This sentence is not clear: the I<O has been used to predict survival (the authors even mention it a few sentences before). What do they mean in this sentence?

·         L97: Missing a dot at the end of the sentence

·         L112: Prof., Mr. (dot at the end)

·         L127: ECOG was already abbreviated before. Moreover, it seems like an important feature, so it should be properly defined here

·         L130: PiPS was already abbreviated before

·         L138: mGPS was already abbreviated before

·         L142: Why were patients instructed to wear the device in the non-dominant arm?

·         L148, L148: the quotation marks here are unnecessary

·         L154: Unnecessary use of simple quotations

·         L158: It is not clear what r24 is. Is that the autocorrelation? Is this another metric? Why is there a slash there? It feels out of place

·         L180: Ranges should be defined using the proper mathematical symbols. How would a correlation of 0.305 be defined?

·         L182: Drop “The”

·         There might be additional minor comments through the manuscript, but as mentioned earlier, since a new version will probably have a major rewrite, I think it does not make sense to further revise the current one.

Author Response

Comments and Suggestions for Authors

In this work, the authors combine data from different sources (actigraphy, sleep journal, clinical data) to predict patient survival in advanced stage cancer patients. This type of modelling could add value to knowledge in the field, since it could potentially provide evidence advocating that a more complete monitoring of a patient could improve his/her outcome.

Unfortunately, there are a few issues that refrain me from suggesting this article for publication. There are two major flaws in the methodology of the proposed analyses. These are explained as follows, together with some other observations. 

Major comments

L97: From reading the introduction, it is not clear to me why the authors decided to focus on patients with advanced cancer.

Authors’ response:

We focus on advanced cancer because accurate prognostication is a crucial aspect of the provision of medical care for patients with advanced cancer, and commonly informs important personal and clinical decisions. This is well recognised in the field (see Hui et al, PMID: 30863893; referenced in our manuscript), and by the first 2 authors of this manuscript, who are specialists in the field of palliative care and care for advanced cancer patients.

The reviewer’s comment made us realise that we hadn’t emphasized the importance for prognostication in advanced cancer. We have therefore now changed the second sentence of the Introduction to emphasize that prognostication is of particular importance for this group of patients. A second reason for focusing on patients with advanced cancer is that the usefulness of the methodology used in this paper (actigraphy) had so far only been explored in a few studies in patients with advanced cancer and in none of these studies had machine learning approaches been used. We had pointed this out in the original submission.

L114: The authors should provide a complete description of the data here, similar to Table 1, explicitly including the descriptive statistics of each feature (e.g., mean and SD for numerical normal features or, probably more realistic, median and IQR for non-normal features; for categorical features the proportion of patients across different groups; the percentage of missing values in each feature). The description of the data used should always go in the M&Ms and never in the Results, since this doesn’t answer any research question, it just provides the reader with information about the used data.

Authors’ response:

Providing the requested information on all the variables considered would in our view be excessive. These variables are mere features used for developing the algorithm. The variables are to the larger extent routine data. We have, however, for each of the numerical variables provided the mean and standard deviation (across all participants) and this information can be found in supplemental table B1.

L118: Moreover, how were patients with locally advanced/metastatic cancer identified? I suppose this was done based on their TNM staging, but this isn’t explicitly mentioned in the text

Authors’ response:

Patients were diagnosed with locally advanced/metastatic cancer according to NHS guidelines which considers TNM staging. We have now added this to the Methods. All patients who met the inclusion criteria were deemed eligible for entry into the study. Potentially eligible patients were identified by the clinical team and approached by a member of the research team and invited to participate in the study.  Please note that any patient referred to the specialist palliative care team (as per definition by the General Medical Council) is in the last one year of life. We have now clarified this in the Methods.

L124: It is not clear how many time points were involved in the data collection. Were they only 2 (day 0 and day 8)? If so, why at those days? Were the data of all patients collected in time?

Authors’ response: 

As reported in ‘Section 2.3. Routine Data collection’, only two time points (day 0 and day 8) were used to collect the routine data.

Data collected on day 0:  patient demographics, information about cancer diagnosis / treatment, information about co-morbidities/medication, assessment of ECOG-PS (by clinician and patient), and completion of the Abbreviated Mental Test Score, the Memorial Symptom Assessment Scale – Short Form (MSAS-SF) and the Global Health Status question from the PiPS-B algorithms. The participant’s pulse was measured (as part of the PiPS), and a venous blood sample was taken to measure haemoglobin, white blood cell count (WBC), neutrophil count, lymphocyte count, platelet count, sodium, potassium, urea, creatinine, albumin, alanine aminotransferase (ALT), alkaline phosphatase (ALP), and C-reactive protein (CRP). 

Data collected on day 8: Repeated assessment of ECOG-PS (by clinician and patient), and completion of the MSAS-SF, the Pittsburgh Sleep Quality Index (PSQI), and a patient acceptability questionnaire. 

Actigraphy and sleep diary data were collected for the seven whole days and nights between day 0 and day 8. These timepoints were selected because we wanted to assess the feasibility to collect data for this duration, observe patient compliance and acceptability of wearing the actigraph device (c.f. PMID: 27707448). The results section L255 reports the number of patients who were fully compliant (per protocol set) and the number of patients who provided at least 72 hours of data (L186).

L159: How were the mean daily activity, and mean activity during daytime wakefulness calculated?

Authors’ response:   

Mean daily activity was calculated as the average number of wrist movements per minute throughout the recording time https://pubmed.ncbi.nlm.nih.gov/19470769/ (Innominato et al., 2009b) and we have now added this to the Methods section. Mean duration of activity during wakefulness, was calculated as the mean activity score (counts/minute) during the time period between two major sleep period intervals - https://pubmed.ncbi.nlm.nih.gov/26273913/ (Ancoli-Israel et al., 2015) and we have now added this to the Methods section.

L160: It is not clear which parameters were calculated automatically from the software and which were calculated manually from the sleep diary. These distinction should be made clear. Moreover, it should be clearly stated how parameters such as sleep latency or sleep efficiency were calculated – or at least how are they defined.

Authors’ response:

We have now stated clearly all the parameters were calculated both automatically from the Actiware software and manually for the sleep diary and have provided a table (Table 1) with the definitions of the sleep parameters used.

Materials and Methods: the organization of this section is a bit messy. Section 2.4 should be a subsection of 2.3.

Authors’ response:

Section 2.4 on Wrist actigraphy and the consensus sleep diary is a crucial aspect of the materials and methods of this manuscript, therefore we have recognised this section as its own section. 

The authors make a distinction between Statistical and Machine Learning methods. This is a gray area and the definition is not crystal clear. Actually, some people might argue that regressions are not part of ML, but more of classical statistics. To avoid this discussion, I suggest that the description of this methods are just grouped under a subsection such as “Data analysis” or so. More importantly, either statistical or ML, this is one of the most important parts of the M&Ms and they should be incorporated into this section as such (and not being reported in a different section)

Authors’ response:

We have amended the heading for Section 4 as – Machine Learning Methods and Data Analysis.

L182: It is not clear to me why did the authors compare survival probability between two groups corresponding to different time points. However, my two biggest issues with the current analysis are the following: The dataset consisted of only 50 patients. This is a very small number for the type of analysis being done. This is particularly worrisome when using models with such a large number of features (66 in one case), which leads to the curse of dimensionality, making it very hard to draw solid conclusions from such a model. In a best case scenario, adding a large amount of patients would solve this issue. However, I know that this is probably very impractical. Thus, I would suggest that the authors drastically reduce the number of features used in their models. They could either pick a few features that have the largest predictive power based on the univariate analysis or they could also engineer features that combine information from some of them.

Authors’ response:

We are well aware of the curse of dimensionality and how it invalidates multiple regression approaches in cases in which there are many predictors relative to the number of observations. Regularised (penalised) regression methods were specifically designed to overcome the curse of dimensionality, and the overfitting problem as we pointed out in the methods section. We validated our approach through leave one out cross validation. We recognise that despite these approaches to minimise the problems of dimensionality and overfitting, our study with its sample size which is appropriate for a feasibility study, nevertheless is limited. We have pointed this out in the Limitations section. Nevertheless, we believe that our study demonstrated the feasibility of constructing a prognostication tool which incorporates features derived from actigraphy and sleep diaries.  

My second largest issue is with the choice of model. The regression model used in this study is inappropriate, since it does not take into account censorship, a crucial issue when doing survival analysis. The authors should have used a Cox Proportional Hazards regression at the very least. If they are interested in the ML aspect, they could further explore the use of Random Survival Forests, Survival Support Vector Machines, or Extreme Gradient Boosting with Cox, to mention a few.

Authors’ response:

We are grateful to the reviewer for pointing out this flaw in our previous analyses. We have now repeated the analyses and accounted for the fact that our survival data are censored. In essence we used glmnet for cox regression for censored data. This has changed some of the results but the essential finding that actigraphy and sleep data are valuable features for prognostication remains unchanged. We have adopted the Methods sections accordingly. 

Given these two important issues, I am afraid that I cannot judge the validity of the drawn conclusions, since they come from a faulty analysis pipeline. Therefore I cannot review the remaining of the manuscript. I encourage the authors to redo their analysis in a proper way and resubmit their manuscript accordingly. I would be happy to review it then. Lastly, I would suggest that if the authors want to compare the performance of different models, they should look into a model-agnostic explainability layer, such as SHAP. This could provide better insights into feature importance for the models, as well as a knowledge representation of what the models do to reach their predictions.

Authors’ response:

We believe that we have corrected our ‘faulty analysis pipeline’ and that the results in our revised manuscript are valid. It is our view that a comparison of the performance of the various models may be useful but it beyond the scope of this feasibility study.

Minor comments - All through the manuscript:

Replace “prognostication” with “prognosis”, it is a more suitable word

Authors’ response:

Prognostication is the most appropriate word for the message that we want to convey in our manuscript. We report on a method of combining tools such as actigraphy, sleep data, bloods e.t.c. using a machine learning approach to improve on the ‘action of’ predicting survival. This is also a recognised word in the Collins and Merriam-Webster dictionaries. Please also see the following reference paper: https://www.ncbi.nlm.nih.gov/pmc/articles/PMC6500464/

Please us the Oxford comma for lists:

Authors’ response:

This has been amended in the revised manuscript.

The authors often use very long sentences, which are hard to read. I have pointed a few in my comments, but not all of them. Please make sure that overall sentences are shortened.

Authors’ response:

We have tried to shorten sentences wherever we can in the revised manuscript.

E.g., and i.e., should always have a comma at the end

Authors’ response:

This has been amended in the revised manuscript.

No need for spaces before/after a slash

Authors’ response:

This has been amended in the revised manuscript.

Abbreviations were often defined twice in the text. They should only be defined the first time they are used, after that, the only the abbreviated version should be used.

Authors’ response:

This has been amended where possible in the revised manuscript.

When reporting physical quantities, there should always be a space between the number and the unit

Authors’ response:

This has been amended in the revised manuscript.

L24: This sentences is too long

Authors’ response:

This has been amended in the revised manuscript.

L30: This sentence could be written better. E.g., “Our findings confirmed the importance of certain well-established survival predictors, as well as identified new ones, in this patient subpopulation”

Authors’ response:

This has been amended in the revised manuscript.

L36: Provide one or two reasons why they remain elusive, since it is not clear how adding actigraphy data would help this issue.

Authors’ response:

Due to word count limitations for the abstract, we are unable to elaborate on this sentence, however the rationale for using actigraphy to improve prognostication has been clearly justified from L80 – L94.

L37: “parameters” repeated in the same sentence

Authors’ response:

The use of ‘parameters’ is intentionally repeated in this sentence as it is in relation to two different tools.

Keywords should be in alphabetical order. Moreover, they should be different from words used in the title. Otherwise, there is no point.

Authors’ response:

This has been amended in the revised manuscript.

 L54: Prognostication is a much broader term and cannot be used as a substitute for survival estimation. If the authors wish to focus or estimating survival, they should refer as such and avoid the more generic term of prognostication (or prognosis)

Authors’ response:

Our focus is on improving the prediction of survival and as no-one knows the exact timing of death with certainty, the prediction is an estimation. The use of ‘prognostication’ in this manuscript has therefore been carefully selected. Please see the following reference paper: https://www.ncbi.nlm.nih.gov/pmc/articles/PMC6500464/.

L55: “and indeed all patients with life-limiting conditions” should be removed. It is obvious and the focus of the paper is cancer patients.

Authors’ response:

This has been amended in the revised manuscript.

L60: This paragraph is very convoluted. It could be easily understood with just the last sentence. Moreover, “that” --> “if”

Authors’ response:

The 3 sentences in this paragraph are making 3 separate points. So we feel this paragraph is justified as it is written. We have substituted “that” with “if” in the last sentence.

L66: It is not clear what the authors mean with objective and subjective items

Authors’ response:

The beginning of the sentence describes that there are a range of prognostic tools for advanced cancer and a reference paper is attached to understand this in further detail: https://www.ncbi.nlm.nih.gov/pmc/articles/PMC6500464/. Our manuscript is combining ‘subjective’ data with data from ‘objective’ tools, which is why these terms have been introduced in the ‘Introduction’ section.

L69: This sentence is too long

Authors’ response:

This has been amended in the revised manuscript.

  L71: The fact that activity and sleep parameters aren’t included in current models should be mentioned after the explanation of why are they important

Authors’ response:

As the beginning paragraphs discuss the importance of prognostication and why it needs to be improved, L71 first introduces the terms related to ‘circadian rhythms’ and that these parameters have not been included in current prognostic tools.. Therefore in keeping with the ‘flow’ of the manuscript, we discuss in further detail the importance of circadian activity and sleep parameters thereafter.

L72: Footnotes with numbers should be favoured

Authors’ response:

This has been amended in the revised manuscript.

L77: survival also in cancer --> cancer survival

Authors’ response:

This has been amended in the revised manuscript.

  L81: Diurnal or circadian was already defined before

Authors’ response:

This has been amended in the revised manuscript.

L86: The 1 points to no footnote

Authors’ response:

This has been amended in the revised manuscript.

  L91: The use of “however” should be favored at the beginning of the sentence

Authors’ response:

This has been amended in the revised manuscript.

L92: This sentence is not clear: the I<O has been used to predict survival (the authors even mention it a few sentences before). What do they mean in this sentence?

Authors’ response:

Whilst the I<O has been found to be a predictor of survival, it has not been used in clinical practice or incorporated in prognostic models to predict survival.

L97: Missing a dot at the end of the sentence

Authors’ response:

This has been amended in the revised manuscript.

L112: Prof., Mr. (dot at the end)

Authors’ response:

This has been amended in the revised manuscript.

L127: ECOG was already abbreviated before. Moreover, it seems like an important feature, so it should be properly defined here

Authors’ response:

This has been amended in the revised manuscript.

L130: PiPS was already abbreviated before

Authors’ response:

This has been amended in the revised manuscript.

L138: mGPS was already abbreviated before

Authors’ response:

This has been amended in the revised manuscript.

L142: Why were patients instructed to wear the device in the non-dominant arm?

Authors’ response:

Application on the non-dominant wrist appears to be the most commonly reported and the best validated approach (c.f. PMID: 12749556, PMID: 26273913). The Actiwatch Spectrum Plus® was therefore applied to the non-dominant wrist of each participant.

L148, L148: the quotation marks here are unnecessary

Authors’ response:

This has been amended in the revised manuscript.

L154: Unnecessary use of simple quotations

Authors’ response:

This has been amended in the revised manuscript.

L158: It is not clear what r24 is. Is that the autocorrelation? Is this another metric? Why is there a slash there? It feels out of place

Authors’ response:

This has been amended for clarity in the revised manuscript.

L180: Ranges should be defined using the proper mathematical symbols. How would a correlation of 0.305 be defined?

Authors’ response:

This has been amended in the revised manuscript.

L182: Drop “The”

Authors’ response:

This has been amended in the revised manuscript.